# Cluster-to-Conquer: A Framework for End-to-End Multi-Instance Learning for Whole Slide Image Classification

**Yash Sharma**[1]                     YS5HD@VIRGINIA.EDU
**Aman Shrivastava**[1]                  AS3EK@VIRGINIA.EDU
**Lubaina Ehsan**[1]                    LE7JG@VIRGINIA.EDU
**Christopher A. Moskaluk**[1]              CAM5P@VIRGINIA.EDU
**Sana Syed**[*1]                  SANA.SYED@VIRGINIA.EDU
**Donald E. Brown**[*1]                  DEB@VIRGINIA.EDU
[1] *University of Virginia, Charlottesville, Virginia, USA*

## Abstract

In recent years, the availability of digitized Whole Slide Images (WSIs) has enabled the use of deep learning-based computer vision techniques for automated disease diagnosis. However, WSIs present unique computational and algorithmic challenges. WSIs are gigapixel-sized ($\sim$100K pixels), making them infeasible to be used directly for training deep neural networks. Also, often only slide-level labels are available for training as detailed annotations are tedious and can be time-consuming for experts. Approaches using multiple-instance learning (MIL) frameworks have been shown to overcome these challenges. Current state-of-the-art approaches divide the learning framework into two decoupled parts: a convolutional neural network (CNN) for encoding the patches followed by an independent aggregation approach for slide-level prediction. In this approach, the aggregation step has no bearing on the representations learned by the CNN encoder. We have proposed an end-to-end framework that clusters the patches from a WSI into $k$-groups, samples $k'$ patches from each group for training, and uses an adaptive attention mechanism for slide level prediction; Cluster-to-Conquer (C2C). We have demonstrated that dividing a WSI into clusters can improve the model training by exposing it to diverse discriminative features extracted from the patches. We regularized the clustering mechanism by introducing a KL-divergence loss between the attention weights of patches in a cluster and the uniform distribution. The framework is optimized end-to-end on slide-level cross-entropy, patch-level cross-entropy, and KL-divergence loss (Implementation: https://github.com/YashSharma/C2C).

**Keywords:** Deep Learning, Multi-Instance Learning, Weak Supervision, Histopathology

## 1. Introduction

Histopathology comprises an essential step in the diagnosis of patients with cancer and gastrointestinal diseases, among others. In recent years, digital pathology has seen an increase in the availability of digitized whole slide images (WSIs) and consequently in the development of novel computational frameworks for computer-aided diagnosis. In particular, a histopathology-based cancer diagnosis has improved significantly via the use of deep learning-based computational frameworks (Bejnordi et al., 2017). These advancements have motivated progress in computational methods for gastrointestinal disease diagnosis (van der Sommen et al. 2020, Syed and Stidham 2020). However, this area poses its unique challenge,

---

* * Co-corresponding Author

including variability in histopathological features across diseases, limited data for training deep learning models, and the extremely high resolution of WSI images ($\sim 100k \times 100k$ pixel). Large WSI sizes make them computationally infeasible to be directly used for the training of deep learning-based techniques. Downsampling images for training leads to loss of relevant cellular and structural details pertinent for diagnosis. Therefore, several recently proposed frameworks follow a two-stage modeling approach where patch-level feature extraction is performed on the WSI followed by an independent aggregation approach to combine the patch level predictions for obtaining patient-level prediction (Campanella et al. 2019, Wang et al. 2016, Wang et al. 2019, Tellez et al. 2019, Lu et al. 2019, Li et al. 2019). This two-stage approach follows a framework known as multiple instance learning (MIL). In MIL, all the instances (patches) coming from a negative slide (non-diseased) comprise a negative label, whereas at least one instance (patch) that comes from a positive slide (diseased) should contain positive class-specific information. Since only a global-image level label is used, this approach is also known as weakly supervised learning (Rony et al., 2019).

Generally, approaches use a patch encoder followed by either a machine learning model or mathematical aggregation such as mean or max pooling for slide-level prediction. The patch encoder can be trained in both supervised or unsupervised settings. With the supervised training set-up, patch-level labels are required for training. Most proposed approaches utilize a MIL-based set-up to train on noisy labels under the mathematical assumption that all the patches from a positive WSI are positive, which may not be true (Campanella et al. 2019, Wang et al. 2019, Hou et al. 2016, Li et al. 2019). Alternatively, these approaches require pathologists to annotate the complete slide at the cellular level for patch-level labels (Wang et al. 2016). Another commonly used approach employs unsupervised methods such as autoencoder or siamese networks for learning patch representation (Tellez et al. 2019, Lu et al. 2019, Li et al. 2020). However, these methods do not guarantee that discriminative features for normal and diseased tissues are being learned. Since the second stage of the model has no control over the learned patterns, it can lead to sub-optimal solutions.

These limitations have increased the interest in an end-to-end training framework using WSIs (Chikontwe et al. 2020, Xie et al. 2020). In this paper, we have proposed an end-to-end approach (C2C) with the following features: (1) Cluster-based sampling for diverse patch selection from a WSI. (2) Attention-based aggregation for slide-level prediction. (3) Inclusion of KL-divergence in the loss for regularizing the intra-cluster variance.

## 2. Related Works

### 2.1. Two-stage model training

The two-stage model training approaches can be further classified into two categories based on whether a supervised learning or an unsupervised learning approach is used to learn patch-level feature representation.

#### 2.1.1. SUPERVISED LEARNING APPROACH

Campanella et al. (2019) proposed a MIL-RNN approach comprising patch level training, top-k instance selection, and RNN-based aggregation for patient-level prediction. Hou et al. (2016) proposed a patch-level classifier that used expectation-maximization for filter-

ing unimportant patches and used an image-level decision fusion model on the histogram of patch level predictions for aggregation. Li et al. (2019) used a multi-scale convolutional layer on top of a pre-trained CNN to capture scale-invariant patterns and top-k pooling to aggregate feature maps for patient-level prediction. Shrivastava et al. (2019) proposed a patch-level classifier with a mean pooling operation for patient-level prediction. Wang et al. (2019) used a patch-classifier and context-aware block selection using spatial contextual information for patch selection before passing the global feature descriptor through a random forest algorithm for patient-level prediction. Lu et al. (2021) proposed a weakly-supervised attention-based learning approach (CLAM), which used a pre-trained encoder to extract all patches representation, followed by attention pooling for WSI classification. Wang et al. (2016) used detailed pathologist annotations for WSIs for patch-level training and a random forest classifier on the extracted geometrical and morphological features for patient-level aggregation.

### 2.1.2. UNSUPERVISED LEARNING APPROACH

Tellez et al. (2019) proposed a neural image compression model that learned patch-level representation using an unsupervised approach followed by spatial consistent aggregation to generate a compressed WSI representation and a standard CNN model for patient-level prediction. Lu et al. (2019) used self-supervised feature learning via contrastive predictive coding and attention-based MIL-pooling for WSI level prediction. Li et al. (2020) used self-supervised contrastive learning for learning patch representation and a dual-stream MIL network for aggregation. Zhu et al. (2017), Yao et al. (2020), and Muhammad et al. (2019) have demonstrated the use of unsupervised learning for discriminative feature representation learning followed by clustering-based sampling for survival analysis tasks.

## 2.2. End-to-End training

In this approach, previous works have attempted to model WSI classification in an end-to-end framework instead of a two-stage approach. Chikontwe et al. (2020) proposed a center embedding approach that employed the joint learning of instance-level and bag-level classifiers and used a center loss for performing end-to-end training with top-$k$ instance sampling. Xie et al. (2020) proposed an end-to-end part learning approach that divided patches from a WSI into $k$ parts based on global clustering centroids and sampled $k$ tiles in each training epoch for end-to-end training. Ilse et al. (2018) proposed the popular neural network-based permutation-invariant aggregation operator that corresponds to the attention mechanism. They demonstrated the efficacy of their approach for identifying the tissue areas indicative of malignancy in breast and colon cancer datasets.

## 3. Methods

### 3.1. Problem Background

For digital pathology classification problems, WSIs ($W$) of patients are available along with their disease labels. Typically, a WSI is in dimensions ranging from $50k \times 50k$ to $100k \times 100k$ pixels, making it computationally infeasible for being directly used for training. Hence, using the Otsu thresholding approach and sliding window approach, patches containing

substantial tissue area ($> 50\%$) of desirable size are extracted. Given a WSI $W$ (bag) with label $y$, we extract $w_1, w_2, w_3, ..., w_n$ patches (instances) from it for training. As we approach the classification problem with the MIL framework, positive bags include at least one diseased patch (instance) while negative bags contain all healthy patches (instances).

To this end, we have developed a convolutional neural network framework, C2C, using: (1) cluster-based sampling method for sampling $N'$ instances from a bag of $N$ instances, (2) end-to-end training using attention aggregation, and (3) inclusion of KL-divergence loss in the clustering set-up for regularizing the attention distribution within a cluster.

## 3.2. Attention-based Aggregation

We used the weighted-average aggregation approach proposed in Ilse et al. (2018) for aggregating the patch-level representation to obtain WSI-level representations. This flexible and adaptive pooling approach uses a two-layered neural network to compute weights for each instance in the bag. Let $H_W = h_1, h_2, \ldots, h_N$ be the $l$-dimension representations of $N$ patches coming from the WSI ($W$), then

$$\mathbf{z} = \sum_{n=1}^{N} a_n \mathbf{h}_n$$

where:

$$a_n = \frac{\exp\left\{\mathbf{v_2}^\top \tanh\left(\mathbf{v_1} \mathbf{h}_n^\top\right)\right\}}{\sum_{j=1}^{N} \exp\left\{\mathbf{v_2}^\top \tanh\left(\mathbf{v_1} \mathbf{h}_j^\top\right)\right\}}$$

where $\mathbf{z}$ denotes the aggregated representation of the WSI, $\mathbf{v_1}$ and $\mathbf{v_2}$ are parameters, and $\mathbf{a_n}$ is attention weight corresponding to $n^{th}$ patch.

## 3.3. Cluster-based Sampling Approach

To accommodate for end-to-end training, we deploy a local cluster-based sampling approach for sampling $N'$ patches from a bag of $N$ patches linked to a WSI ($W$). The cluster-based sampling approach exposes the model to diverse discriminative patches from WSIs. Local clustering (clustering patches from a single WSI) is preferred over the global clustering approaches (clustering patches from all the WSI) as the latter is susceptible to creating clusters based on the visual biases such as variation to staining or scanning procedure instead of medically relevant features.

A patch-level encoder, $G_e(\mathbf{x}; \Theta_e) : \mathbf{x} \rightarrow \mathbf{h}$, maps all patches to $l$-dimensional embeddings where $\Theta_e$ is the set of trainable parameters. These parameters are frozen during the clustering step. $K$-means clustering is performed independently for each of the WSIs using all the patches for dividing and grouping WSI patches into $k$ different buckets. Equal $k'$ patches are sampled from each of the $k$ clusters, keeping the maximum number of patches sampled from a WSI to 64 ($N' \geq k' \times k$). The maximum number of patches sampled from each cluster is kept at 64 to accommodate for computational limitation. As the representations get richer, we hypothesize that these $N'$ randomly sampled instances approximate the representation of a WSI with $N$ patches.

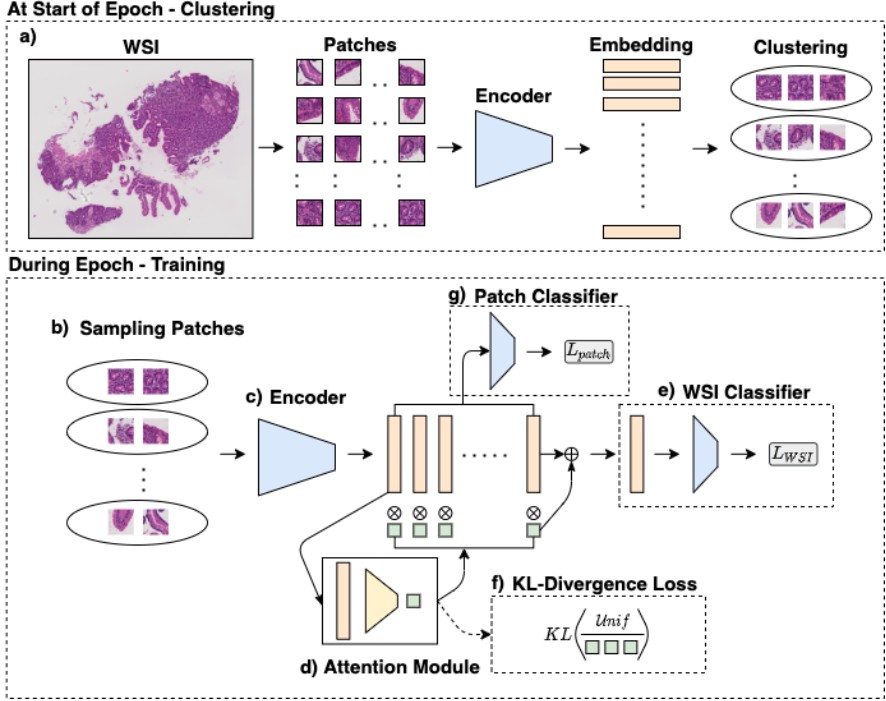

Figure 1: Overview of the proposed Cluster-to-Conquer (C2C) framework. a) At the start of each epoch, representation of all the patches of one WSI are extracted. K-means clustering is performed for segregating patches coming from the WSI into $k$-buckets. b) $k'$ patches are sampled from each of the $k$ clusters for end-to-end training. c) Using encoder, representation is generated for all the patches. d) Patch representations are passed through a 2-layer fully connected attention module for weight calculation. e) Using the weighted aggregation, representation for the WSI is generated and passed to the WSI classifier. f) Attention weights corresponding to patches coming from the same cluster are passed to the KL-divergence module to penalize the high intra-cluster attention variance. g) Patch representations are passed to instance classifier for training with weak supervision.

### 3.4. End-to-End Learning

The sampled instances $h_1, h_2, \ldots, h_{N'}$ are passed through attention aggregation module $G_a(h_{i=1}^{N'}; \Theta_a) : \mathbf{h} \to \mathbf{z}$ where $\Theta_a$ are the trainable parameters for generating the aggregated representation of the WSI. Using the aggregated representation of the WSI and representation of patches (instances), end-to-end training is performed using cross-entropy and KL-divergence loss. The aggregated representation is passed through $G_y : \mathbf{z} \to \mathbf{y}$ to obtain WSI prediction probability and the instance representation are passed through $G_{y'} : \mathbf{h} \to \mathbf{y'}$ to obtain patches prediction probability. Instance loss is included with weak supervision assumption. All the patches from a diseased tissue are treated as diseased, and all the patches coming from healthy tissue are healthy to compensate for limited size training.

$\Theta_a$, $\Theta_e$, $\Theta_y$ and $\Theta_{y'}$ are trained in each epoch with clustering performed using $\Theta_e$ representation at the start of each epoch for sampling patches. Patches are sampled randomly from each cluster to regularize the model. Along with WSI and patch cross-entropy loss, for each cluster, KL-divergence loss between the patches' attention weight and a uniform distribution is included. The inclusion of KL-divergence loss regularizes the same cluster patches' attention distribution and allows the attention module to weight all the positive class patches uniformly. The aggregated loss can be written as:

$$L(G_y, G_{y'}, G_a, G_e) = \alpha * L_{WSI} + \beta * L_{Patch} + \gamma * L_{KLD}$$

where $\alpha$, $\beta$, and $\gamma$ balance the importance of different losses.

### 3.5. Architecture and Hardware

For the base architecture, we used ResNet-18 (He et al., 2015) with a combination of the linear layer to reduce 512 flattened representation to $l$ (64 in our case). Clustering and attention pooling were performed on this $l$-dimension representation. Clustering was performed using the $k$-means algorithm with $l2$-normalization. The model was implemented with PyTorch and trained on a single RTX2080 GPU. The framework was trained end-to-end with Adam optimizer with a batch size of 1 and a learning rate of $1e-4$ for 30 epochs. Empirically, $\alpha = 1$, $\beta = 0.01$ and $\gamma = 0.1$ for loss hyperparameters demonstrated best performance. We experimented with different $k$. $K = 8$ had the best performance (Refer to Appendix for $k$ comparison). All the patches were used with the attention aggregation module for computing WSI representation during the inference stage before passing them through the classifier layer for final prediction.

## 4. Experiment and Results

### 4.1. Data Description

We demonstrated our approach and compared it with standard approaches on a gastrointestinal dataset containing 413 high-resolution WSIs obtained from digitizing 124 H&E stained duodenal biopsy slides (where each slide could have one or more biopsy images) at $40\times$ magnification. The biopsies were from children who went endoscopy procedures at the University of Virginia Hospital. There were 63 children with Celiac Disease (CD) and 61 healthy children (with histologically normal biopsies). A 65%-15%-20% split was used to split data for training, validation, and testing. Patches with at least 50% tissue area of size $512 \times 512$ were extracted from each WSI. We have also reported our performance on the publicly available CAMELYON16 dataset for breast cancer metastasis detection and have compared it to the fully-supervised approaches. For the CAMELYON16 dataset, patches of size $512 \times 512$ at $10\times$ magnification were extracted with at least 50% tissue area. For studying the effect of the KL-divergence loss, we experimented on a widely used MNIST-bags MIL dataset. We created 400 bags of MNIST instances for training and 100 bags for testing. We defined a bag as positive if it contained either the numbers 8 or 9 (details provided in Appendix). We used the LeNet5 (LeCun et al., 1998) model as encoder with our proposed changes for demonstrating the effect of KL-divergence loss on training.

Table 1: Evaluation of our proposed method (C2C) against standard approaches on Gastrointestinal Data for classifying Celiac vs. Normal. Avg. of 3 runs are reported.

| Method | Accuracy | Precision | Recall | F1-Score |
|---|---|---|---|---|
| Campanella-MIL | 82.8 | 94.9 | 74.5 | 83.5 |
| Campanella-MIL RNN | 74.7 | 75.4 | 84.3 | 79.6 |
| Two-Stage Mean | 81.6 | 87.3 | 80.3 | 83.7 |
| C2C (w WSI Loss) | 81.6 | 80.7 | 90.1 | 85.2 |
| C2C (w WSI+KLD Loss) | 83.9 | 84.9 | 86.3 | 85.4 |
| C2C (w WSI+Patch Loss) | 85.1 | 86.5 | 88.2 | 87.4 |
| C2C (w WSI+Patch+KLD Loss) | **86.2** | 85.5 | 92.2 | 88.7 |

### 4.2. Evaluation

We compared our proposed approach with two approaches: (1) The two-stage state-of-the-approach proposed in Campanella et al. (2019). (Campanella-MIL and Campanella-MIL RNN)[1] and (2) the two-stage mean pooling approach proposed in Shrivastava et al. (2019) (Two-Stage Mean Pooling)[2]. The Campanella-MIL approach used ResNet-34 as the backbone architecture, and the Two-Stage Mean Pooling approach used ResNet-50 as the backbone architecture.

Table 1 demonstrates the performance of the C2C method. We observed that even with a relatively weaker ResNet backbone among comparison methods, C2C performs better than other approaches. We attributed this to the synergy between the aggregation and encoding module that the C2C framework can achieve using end-to-end training of an encoder and aggregation module. In the two-stage modeling approach, the encoder is decoupled from the aggregation module, leading to sub-optimal learning for the classification task.

Additionally, to verify if C2C consistently created clusters and assigned high attention weights to relevant patches. We randomly sampled Celiac Disease WSIs from our test set and got the patches with their cluster allotment and attention weights reviewed by a medical expert. The high importance patches highlighted damaged surface epithelium, which indicates tissue inflammation present in Celiac Disease (Liu et al., 2020) and intraepithelial lymphocytes in the surface epithelium, which is a histopathologic feature used for Celiac Disease diagnosis (Oberhuber, 2000). Medical expert review has been explained in detail in the Appendix with figure Figure 3 showing sampled patches from each cluster in descending order of their attention weights.

In the CAMELYON16 dataset, by training the model only on slide-level labels, C2C achieved a strong performance of **0.9112** ROC-AUC score on the test dataset. This performance would have ranked second on the classification portion of CAMELYON16 challenge (best model by Wang et al. (2016) achieved an AUC score of 0.9223) and seventh on the open leaderboard(Bejnordi et al., 2017). We have reported a competitive performance compared

---

1. https://github.com/MSKCC-Computational-Pathology/MIL-nature-medicine-2019

2. https://github.com/GutIntelligenceLab/histo_visual_recog

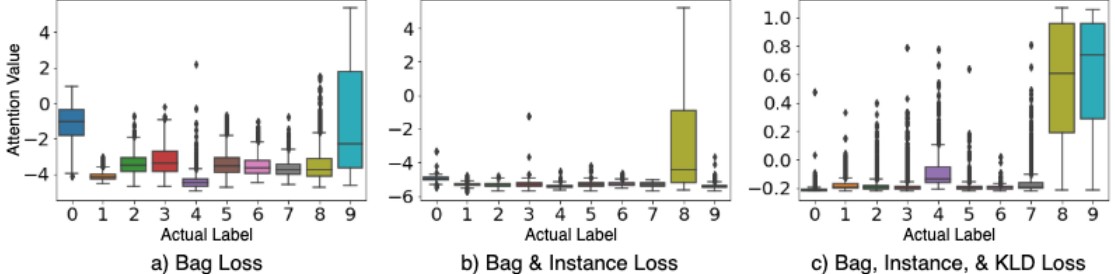

Figure 2: Inclusion of KL-divergence regularizes the attention value corresponding to the positive instance classes - 8 and 9. When the model was trained with a) Bag loss or b) Bag and Instance loss, the attention module randomly selected one of the positive instance classes and gave it the highest attention value. In contrast, when the model was trained with Bag, Instance, and KL-divergence loss, the attention module gave equal importance to both the positive instance classes - 8 and 9.

to fully supervised techniques trained using detailed pathologist annotations [3]. Details of our model and examples of high attentive patches overlaid on the tumor maps are shared in the Appendix section. We observed that C2C could accurately identify the patches with tumor regions and assign them higher attention weights.

Using MNIST-bag data, we demonstrated the value of including KL-divergence in our approach. The inclusion of KL-divergence regularizes the high instance variance of attention distribution observed in similar positive instances, Figure 2. All models reported in the Figure 2 quickly converged to higher accuracy as expected in the MNIST-bag experiment. However, without KL-divergence loss, attention weights for positive instance classes 8 and 9 were highly variable in different bags. We observed that by including KL-divergence loss, attention weight became more uniform for both the positive instance classes.

## 5. Conclusion

In this paper, we proposed an end-to-end Whole Slide Image (WSI) Classification framework using clustering-based sampling technique, adaptive attention module, and KL-divergence loss. We demonstrated strong performance of the proposed framework for celiac disease and breast cancer classification. C2C is able to achieve comparable performance to fully supervised methods trained using detailed pathologist annotation. This highlights the power of building strong MIL frameworks. More importantly, clusters with high attention maps in breast cancer overlap with pathologist-annotated tumor areas, and top clusters of celiac disease match the patterns deemed important by medical experts for diagnosing celiac disease. In our future work, we will explore the performance of the C2C framework on multi-class and sub-type classification problems.

---

3. https://camelyon16.grand-challenge.org/Results/

## Acknowledgments

This work was supported by NIDDK of the National Institutes of Health under award number K23DK117061-01A1.

We would like to thank members of the Gastroenterology Data Science Lab [4] for their valuable comments on the architecture and experiment, and UVA Research Computing group [5] for their support with high performance computing environment.

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

## Appendix A. Example and UMAP Plot of WSI

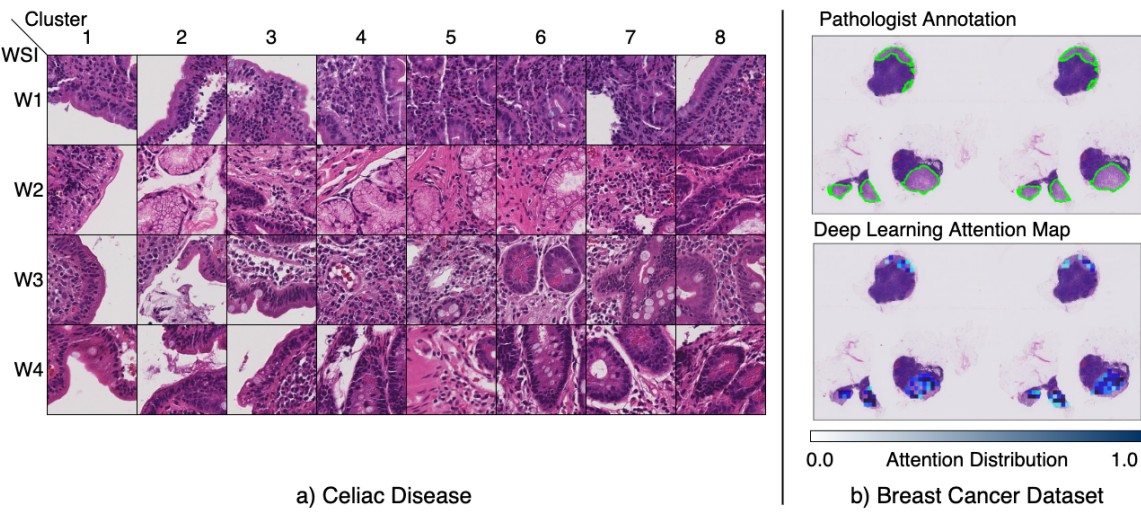

Figure 3: a) Gastrointestinal Dataset - Patches sampled from clusters of different whole slide images in decreasing order of attention importance for detecting Celiac disease. b) CAMELYON Dataset - Top figure contains the actual tumor regions annotated by the pathologists, and the bottom figure contains the patches assigned high attention importance by our model.

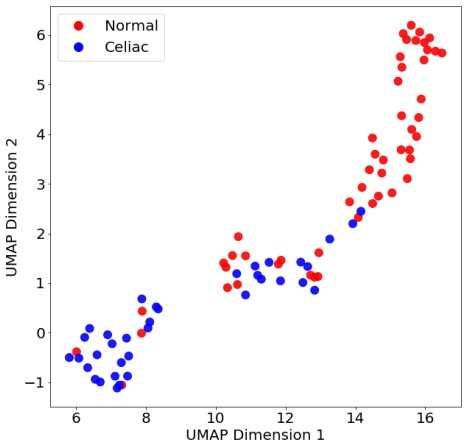

Figure 4: WSI embedding representation of Celiac and Normal biopsies in the test dataset.

## Appendix B. MNIST Bag Set-Up

To investigate the impact of the inclusion of patch loss and KL-divergence loss to attention map distribution, we used the well-known MNIST image bag dataset proposed in Ilse et al. (2018). In MNIST, a bag is made up of $28 \times 28$ grayscale images of random numbers. The number of images in a bag is Gaussian-distributed, and the closest integer value is taken. A bag is given a positive label if it contains either an '8' or a '9'. For all experiments, a LeNet5 model is used (LeCun et al., 1998) as an encoder with cluster sampling technique and adaptive attention aggregation for bag prediction. In the experiments, we use a bag of mean size 400 with a variance of 100 for creating problems similar to the WSI-modeling scenario. We compared how the inclusion of instance and KL-divergence loss with bag loss changed the distribution of attention weights. All of our experiments quickly converged to an accuracy of 100% with a different distribution of attention weights. We report that the inclusion KL-divergence loss regularizes the attention weight distribution for positive instance classes.

## Appendix C. Impact of Different Parameters on the Performance of Celiac Disease vs. Histologically Normal Dataset and Inference Time

Table 2: Performance of the model on test dataset with different number of clusters. For training, we sample at max 64 ($N'$) patches per WSI.

| Number of Clusters | Accuracy | Precision | Recall | F1-Score |
|---|---|---|---|---|
| $k=4$ | 81.6 | 80.7 | 90.2 | 85.2 |
| $k=6$ | 78.6 | 76.63 | 90.2 | 82.3 |
| $k=8$ | 86.2 | 85.5 | 92.2 | 88.7 |
| $k=10$ | 79.3 | 81.13 | 84.31 | 82.7 |

Table 3: Performance of the model on test dataset with different sampling strategies. For training, we sampled at max 64 ($N'$) patches per WSI.

| Sampling Strategy | Accuracy | Precision | Recall | F1-Score |
|---|---|---|---|---|
| Cluster | 86.2 | 85.5 | 92.2 | 88.7 |
| Top-K | 82.75 | 87.2 | 82.35 | 84.84 |

Table 4: Sensitivity analysis of gamma (KL-divergence loss weight) on performance. For training, we sampled at max 64 ($N'$) patches per WSI.

| KL-Divergence Loss Weight | Accuracy | Precision | Recall | F1-Score |
|---|---|---|---|---|
| $\gamma = 1$ | 81.6 | 87.2 | 80.4 | 83.7 |
| $\gamma = 0.1$ | 86.2 | 85.5 | 92.2 | 88.7 |
| $\gamma = 0.01$ | 83.9 | 84.9 | 88.2 | 86.5 |

Table 5: Performance of the model on test dataset with different pooling strategy. For training, we sampled at max 64 ($N'$) patches per WSI.

| Pooling Strategy | Accuracy | Precision | Recall | F1-Score |
|---|---|---|---|---|
| Mean Pooling | 85.05 | 85.1 | 90.1 | 87.6 |
| Attention Pooling | 86.2 | 85.5 | 92.2 | 88.7 |

Table 6: Inference time per WSI in test dataset.

| Approach | Inference time (sec) |
|---|---|
| C2C | 2.2 |
| Campanella-MIL | 1.8 |
| Campanella-MIL RNN | 2.1 |
| Two-Stage Mean | 2.5 |

## Appendix D. Medical Expert Qualitative Review

| WSI\Cluster | Cluster 1 | Cluster 2 | Cluster 3 | Cluster 4 | Cluster 5 | Cluster 6 | Cluster 7 | Cluster 8 |
|---|---|---|---|---|---|---|---|---|
| **WSI-1** | damaged surface epithelium and connective tissue | damaged tissue and brunner glands | connective tissue and brunner glands | connective tissue and brunner glands | crypt cross sections with crowded epithelial nuclei and brunner glands | connective tissue and brunner glands | crowded nuclei in lamina propria and connective tissue | crypt cross sections with crowded epithelial nuclei |
| **WSI-2** | damaged surface epithelium and surface epithelium with intraepithelial lymphocytes | crowded nuclei in lamina propria and epithelium and damaged tissue surface - which maybe an artifact or due to tissue inflammation | crowded nuclei in lamina propria | crowded nuclei in lamina propria | crowded nuclei in lamina propria with crypt cross sections | crowded nuclei in lamina propria with crypt cross sections | inconclusive: has too many features: crowded nuclei in lamina propria, damaged surface epithelium, and crypt cross sections | crowded nuclei in lamina propria |
| **WSI-3** | surface epithelium with intraepithelial lymphocytes | damaged surface epithelium and tissue surface with crowded nuclei | artefactual tissue separation with crowded nuclei | crowded nuclei and connective tissue in lamina propria | crowded nuclei and connective tissue in lamina propria | crypt cross sections with prominent enteroendocrine cells and paneth cells | crowded nuclei in lamina propria and crypt cross sections | crowded nuclei in lamina propria and crypt cross sections |
| **WSI-4** | surface epithelium with intraepithelial lymphocytes | Less columnar surface epithelium with intraepithelial lymphocytes | Less columnar surface epithelium with intraepithelial lymphocytes and connective tissue | crypt cross sections with crowded epithelial nuclei | areas of connective tissue within and outside lamina propria | crypt cross sections with crowded epithelial nuclei | crypt cross sections with prominent paneth cells | crypt cross sections with prominent paneth cells |

In the table above, we present a qualitative assessment of the patch clusters by a medical expert. Cluster 1 was of the highest importance, while Cluster 8 was the lowest. Top clusters (Cluster 1) for the WSIs included histopathologic features important for celiac disease diagnosis along with the assessment of tissue inflammation and celiac disease severity. These include intraepithelial lymphocytes, diagnosis and severity assessment as per modified Marsh-Oberhuber classification (Oberhuber, 2000), and damaged surface epithelium indicative of tissue injury due to inflammation (Liu et al., 2020). Other histopathologic features identified were brunner glands and less columnar epithelium comparison to histologically normal duodenum. These findings are supported by literature to be present in intestinal inflammation due to enteropathies such as celiac disease where brunner gland hyperplasia (Liu et al., 2020) and loss of columnar epithelium is noted (Dickson et al., 2006). Enteroendocrine cells were also noted in Cluster 6 of WSI 3 that have been reported to be present in duodenal biopsies of patients with refractory celiac disease (Di Sabatino et al.,

2014). These findings show that our method utilized medically relevant histopathologic features to cluster patches from WSIs.

## Appendix E. CAMELYON16 Model and Examples

For the CAMELYON16 model, we used ResNet 18 as the backbone encoder with $\alpha = 0.01$ for WSI loss, $\beta = 1$ for Patch loss, and $\gamma = 0.01$ for KL-Divergence loss. As suggested in Raghu et al., we reinitialized running mean to unit distribution and running std to zero along with increasing momentum to 0.9 to tackle for small batch-size. The model was trained with 85%-15% training-validation split for 30 epochs with Adam optimizer and a learning rate of $1e-4$. The number of clusters ($k = 8$) and patches sampled per cluster ($k' = 8$) was kept similar to our experiment with the celiac disease dataset.

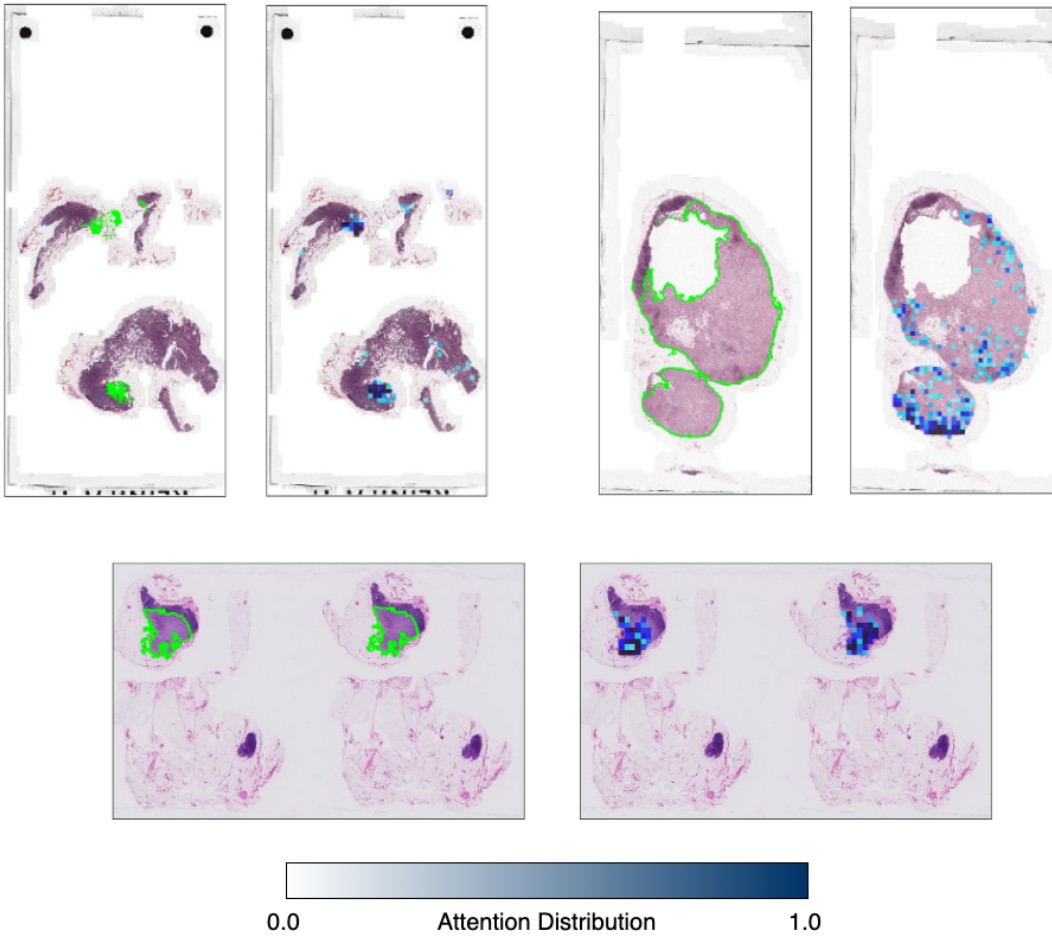

Figure 5: Cancerous WSIs from CAMELYON16 data with pathologist-annotated tumor area and deep learning assigned attention distribution. For each WSI, left figure shows the pathologist annotation and right shows the deep learning results.

## Appendix F. Limitations

In this section, we present the limitations we observed in our approach. A batch size of 1 WSI is used for training, leading to unstable peaks in training depending on the normalization strategy adopted. In the proposed MIL framework, each batch contains 64 patches from a WSI; hence, the batch normalization is typically used for WSI normalization in each iteration. We used momentum tuning and reinitialize running *mean* and *std* to stabilize the training as proposed in Raghu et al.. Clustering is also a time-intensive step and can slow down the training. Xie et al. (2020) randomly sampled 10% of the slides in their global clustering approach for handling the large number of patches coming from a WSI. We will experiment with similar strategies for optimizing C2C training without impacting the performance in our future work.

