# OpenReview forum: "Cluster-to-Conquer: A Framework for End-to-End Multi-Instance Learning for Whole Slide Image Classification"
_MIDL.io/2021/Conference — MIDL 2021_

### Official Review · AnonReviewer4 · 2021-03-06

**Confidence:** 4
**Preliminary Rating:** 4
**Recommendation:** Oral

**Summary:**

In the paper "Cluster-to-Conquer: A Framework for End-to-End Multi-Instance Learning for Whole Slide Image Classification" the authors proposed and investigate a strategy for end-to-end whole-slide classification. The main assumption is based on the clustering of patches from a single WSI into k-groups following by an adaptive attention mechanism for a slide level classification. The proposed approach was compared with state-of-the-art solutions and evaluated using two various datasets, which allows demonstrating general method possibility.

**Strengths:**

The paper is solid. The method is new, interesting, and well described. The research goal is clear and important. The achieved results are good and significant for the research community. The main benefits are evaluations on two, different, datasets.

**Weaknesses:**

The main weakness of the paper is the lack of details about the processing time. How is this method impacting on time of model training and single slide classification? Is single slide classification faster ?

Other comments
- all symbols used in the equation should be explained
- is not clarified if k' has equal value for each cluster?
- maximum number of patches was established as 64- how this value was established?
- it is useful to present an average time of single slide classification for each method (Table 1)

**Deanonymize Review:**

no

**Detailed Comments:**

- caption used on figures should be readable, now they are too small (figure 1, figure 2, figure 3)
- numbers in the table (Table 1) should be with the same precision

**Justification Of The Preliminary Rating:**

The proposed method is interesting and was evaluated on 2 independent datasets, which shows their robustness and practice application. The authors clearly described the method and strategy and compared achieved results with other methods available in the literature.

**Paper Type:**

methodological development

**Questions To Address In The Rebuttal:**

- is the proposed method publicly available?


**Special Issue:**

yes

---

> ### Author Response · Authors · 2021-03-18
> **Response to Reviewer 4**
>
> Thank you for your valuable feedback and comments!
>
> Kindly find our response to the comments -
>
> - **Time Complexity:** We have included the inference time of all the approaches for single slide classification in the Appendix. For the training step, as highlighted in the limitations in Appendix, we are working on optimizing the clustering process as this can slow down the training process when we train our model on a bigger dataset. One of the potential ideas, as used in Xie et al., is to sample a set of patches in training iteration for each WSI.
> - **Symbols in the equation:** We have updated the manuscript to explain all the equation's symbols clearly.
> - **Maximum number of patches:** We chose 64 as it is approximately the maximum number of 512x512 pixel patches we can keep in our batch for training the model without getting memory issues. We have added details in the experiment section highlighting this reason.
> - **Figure and their captions:** We apologize for the low-quality images and have updated images and captions to support our framework.
> - **Same precision results:** Thank you for pointing this out. We have updated our results in Table 1.
> - **Method publicly available:** Yes, we have open-sourced our approach here - https://github.com/YashSharma/C2C.

---

### Official Review · AnonReviewer1 · 2021-03-08

**Confidence:** 4
**Preliminary Rating:** 3
**Recommendation:** Oral
**Final Rating:** 4

**Summary:**

This paper presents an approach for WSI classification in which there are two modules, the first one clustering patch-level encodings by k-means, and the second one learning the best encodings to classify the slide based on optimal weights for each cluster, with extra supervision at the patch and slide level. A regularization term to avoid the learned attention weights from becoming too sharp is also presented, based on making them close to a uniform distribution by means of the KL loss. The method seems to be properly validated on a private dataset for celiac disease, and on the public CAMELYON16 dataset.

**Strengths:**

The paper is clearly written and reads quickly, which is appreciated. The review on previous work is very nice, with the division between end-to-end learning and two stage methods helping the reader to contextualize the focus of this work. The experiments are comprehensive, with some ablation studies included. A nice review of an expert on the clinical meaning of the regions to which the method attends is provided also in the appendix.

**Weaknesses:**

Not considering some technical aspects that I will discuss below, what I find a bit annoying in this paper is the relatively lower effort invested in creating the figures and tables that illustrate it. Figure 3 contains some extremely small bar plots with hardly legible labels. Indeed, legends and labels are almost unreadable also in the other Figures.

As a curiosity, it is never mentioned the number of classes in each problem. Is it always binary classification, both in the celiac disease task and in CAMELYON? If the number of classes is different, shouldn't that affect the number of clusters selected?

"As the representations get richer, we hypothesize that these N' randomly sampled instances approximate the representation of a WSI with N patches." I find this to be relatively speculative, and I believe this could maybe verified with some experiment. If not on the WSI scenario, on the MNIST problem? It would be very nice to see that the representations indeed get closer in each case, numerically.


**Deanonymize Review:**

no

**Detailed Comments:**

It seems to me from Table 1 that the KL regularization plays a central role in the success of the presented approach. Apparently, without imposing it, the model would give too much weight to too few patches and ignore much relevant information, which is confirmed by the MNIST experiment and figure 3. Maybe the authors could provide a small sensitivity analysis on the impact of gamma as a hyperparameter, since it seems to be so important? If varing gamma leads to catastrophical results, that would be a concerning aspect of the method that should be discussed in the conclusions.

By the way, what some peope call MIL learning in this context (at the patch level) could be probably connected to what other people call weakly supervised learning (this time providing info at the path and maybe also pixel level). I understand that adding this family of methods to the literature review would make it too long, but maybe the authors may want to point the reader to some review on this similar perspective of the same problem. I could suggest this one: https://arxiv.org/abs/1909.03354

**Final Rating Justification:**

The paper was already in good shape before the rebuttal. It seems that the authors have made a noticeable effort in answering all reviewer's comments. I am happy to increase my recommendation, not only mine. I appreciate the provided sensitivity analysis that was missing.

**Justification Of The Preliminary Rating:**

The paper is well-written, and the method looks well motivated and consistenly developed. The paper could be improved in some aspects, particularly the graphical content is quite poor. I think with some further work this could be made a strong accept.

**Paper Type:**

methodological development

**Questions To Address In The Rebuttal:**

Mostly what I mentioned above. Please fix the figures, and the sensitivity analysis on gamma would also be muh appreciated.

**Special Issue:**

no

---

> ### Author Response · Authors · 2021-03-18
> **Response to Reviewer 1**
>
> Thank you for your valuable feedback and comments.
>
> Kindly find below our response to raised concerns:
>
> - **Figure:** We apologize for the low-quality figure and have updated the figures to communicate our results better.
> -**Binary vs. Multi-class classification:** Yes, both of the problems used in our paper for validation are binary classification problems. We have added details to highlight this.
> -**N’ patches approximating WSI with N patches:** We would like to highlight that based on medical reviews, we can confirm that the model in our framework is learning relevant patterns. In MNIST tasks, we can observe that our proposed losses in the model are giving attention to all the positive class instances. To establish that our model is learning the discriminative representation of WSI, we have added UMAP plots in the Appendix to compare normal vs. disease WSI representation.
> -**Sensitivity analysis of gamma:**  We have added the table comparing performance for different values of gamma in the duodenal biopsy task in the Appendix. To further track the performance variation that can occur due to random sampling of patches and different combinations of weights, we ran all of the experiments reported in Tables three times and have reported the average performance. The updated average performance numbers highlight that KLD loss alone with WSI loss doesn’t give a big jump; instead, it is the inclusion of patch loss with WSI and KLD loss that leads to the jump. This behavior was further confirmed in the CAMELYON experiment.
> -**Literature Review:** Thank you for pointing us in this direction. In our introduction, we have added a connection between multiple instance learning with weak supervised learning and references for redirecting interested readers to this area.
> -**Further work:** We are working on extending the approaches proposed in this paper for multi-class classification problems and experimenting with different backbone architecture for comparing imagenet pre-trained network with self-supervised learning pretrained network trained on histopathology images (https://arxiv.org/pdf/2011.13971.pdf) and multi-task classification network pretrained for multiple histopathology tasks (https://arxiv.org/pdf/2005.02561.pdf).

---

### Official Review · AnonReviewer3 · 2021-03-08

**Confidence:** 5
**Preliminary Rating:** 2
**Final Rating:** 3

**Summary:**

This paper presents a method for end-to-end training of CNN to predict a single label at whole-slide image level in digital pathology images.
The method is based on attention-based aggregation as proposed by Ilse et al., a cluster-based sampling strategy and an attention model have been added.
Performance is reported on a dataset of 124 slides of duodenal biopsies for the diagnosis of celiac disease and Camelyon16, where this method would have scored 2nd in the official challenge and within the top-10 in the open (post-challenge) leaderboard.

**Strengths:**

* This method presents an end-to-end solution to whole-slide image classification, which is a relevant context in computational pathology research, where manual annotations are often scarce and time-consuming.
* This method outperforms some state-of-the-art methods like the MIL method of Campanella et al. However, it is not surprising that the performance is superior when compared to Campanella-MIL, as the authors of Campanella et al. show that their method does not work well when the training set size is small.

**Weaknesses:**

* The authors should clearly show the difference between their method and the CLAM method (preprint in April 2020, https://arxiv.org/abs/2004.09666, recently published: https://doi.org/10.1038/s41551-020-00682-w), which also uses clustering + attention methods. It seems that the main contribution of this paper over CLAM is the fact that it can be trained end-to-end, while CLAM relies on a pre-trained encoder. A clustering method was also used in Xie et al., therefore the authors should clearly show what the contribution of this work is when compared to existing work.
* Figure 2b is too small.
* Not clear if the results in Table 1 are from Camelyon16 or from the duodenal biopsies dataset
* What label is used in the Lpatch loss, the WSI label? This seems to have little effect on the final performance, it would have been good to check the combination C2C with WSI+KLD loss without patch loss, to make a complete overview of the contributions
* A comparison with the method of Ilse et al. should be reported as well, as the method presented in this paper uses the same attention-based aggregation strategy.

**Deanonymize Review:**

no

**Final Rating Justification:**

I would like to thank the authors for their comments and for clarifying some aspects about the novelty of their work.
It would have been interesting to see in the paper a discussion and comparison between the CLAM method with the proposed method, as it seems that the proposed method has some advantages compared to CLAM; same for a comparison with the method of Xie et al.

**Justification Of The Preliminary Rating:**

The novel terms presented in this paper, i.e., clustering and attention mechanisms contribute to outperform some existing methods on the same datasets. However, the authors should consider more recent work, such as the CLAM method, and clarify the novelty of this paper when compared to other work.

**Paper Type:**

both

**Special Issue:**

no

---

> ### Author Response · Authors · 2021-03-18
> **Response to Reviewer 3 (1/2)**
>
> Thank you for your valuable feedback and comments!
>
> Kindly find our response to the comments-
>
> - **Comparison with CLAM:** Thanks for pointing us to this amazing work. In the paper, the authors have demonstrated strong interpretable data-efficient performance using the proposed CLAM approach. We want to highlight the following differences in our proposed approach from CLAM:
>     * CLAM is a 2-stage approach as it uses a frozen backbone encoder for extracting features and then trains an attention-based aggregation module for WSI prediction. In comparison, our framework performs end-to-end training of backbone encoder and uses an attention-based aggregation module for WSI prediction.
>     * Since the CLAM method uses a frozen backbone, they can extract feature representation for all the patches and use it for training. In our case, for achieving end-to-end training, we are using k-means clustering for sampling diverse patches. As demonstrated in our results section, end-to-end training can achieve better performance with smaller networks. In CLAM, authors have used imagenet pre-trained ResNet50 for extracting representation for patches, whereas we fine-tune ResNet18 in our framework and achieve strong performance. Empirically, CLAM paper shows that ResNet50 pretrained on imagenet identifies relevant patterns. However, considering distributional variation between imagenet data and any histopathology data, it is better to fine-tune backbone models on histopathology data for domain adoption before using it for feature extraction.
>     * The clustering used in CLAM is not exactly clustering in the traditional sense. Instead, they have used strongly attended patches as positive class and weakly attended patches as the negative class for training patch level classifiers to increase the separation between positive and negative patches and provide more signals to the model training.
>     * In the paper, for getting access to more WSIs, the authors have combined CAMELYON16 and CAMELYON17 data for training. We can use their reported numbers on the CAMELYON dataset for a proxy comparison. Authors have highlighted for achieving 0.9 AUC in the lymph node classification (CAMELYON) task, they need at least 289 WSIs. In contrast, in our approach, for CAMELYON16 experimentation, we are using 231 WSIs for training and 39 WSIs for validation and achieving a 0.9 AUC score. This is not an exact comparison but serves as a good proxy.
>     * We hope this highlights our unique contribution from the CLAM method. We have updated the manuscript to include CLAM in our related work section.
> - **Xi et al. Clustering method:** We agree that the clustering method was also employed in the Xie et al. approach, which we have highlighted in our related works. But we would like to point out that in Xie et al., they have used a global clustering approach, whereas we used a local clustering approach. We argue that considering deep learning models are sensitive to stain characteristics like stain color, in global clustering, all the patches from a WSI can get clustered into the same cluster limiting the diverse sampling required for training classification algorithms. Hence, demanding a bigger dataset for training. In contrast, local clustering allows us to sample diverse patches from a WSI and train better on even small datasets.
> - **Figure:** We apologize for the low-quality figure and have updated all the figures. We have included a few more examples for CAMELYON in the appendix section to demonstrate the accuracy of attention distribution.
> - **Caption:** Results in Table 1 are for Duodenal biopsy. We have updated the caption of the figure to highlight this.
> - **Patch loss:** We agree that adding C2C with WSI+KLD loss will give a complete overview of our approach and have updated the table to include C2C with WSI+KLD loss. To further compare the performance variation that can occur due to random sampling of patches and different combinations of weights, we ran all of the experiments reported in Table 1 three times and have reported the average performance. We used the same label as the WSI label for training patch classifier. The updated average performance numbers highlight that KLD loss alone with WSI loss doesn’t give a big jump; instead, it is the inclusion of patch loss with WSI and KLD loss that leads to the jump. Our motivation for including patch loss is-
>     * We hypothesize for a small dataset, including patch loss, stabilizes the training as demonstrated in CLAM and Chikontwe et al. (https://link.springer.com/chapter/10.1007/978-3-030-59722-1_50).
>     * For CAMELYON16, the weight of patch loss is 1, and without a patch loss model wasn't giving us competitive performance.

---

> ### Author Response · Authors · 2021-03-18
> **Response to Reviewer 3 (2/2)**
>
>
> - **Ilse et al. comparison:** Computationally, it is not possible to run the approach proposed in Ilse et al. as in the training stage, all the patches are used for training the proposed end-to-end framework. In the paper, the authors have demonstrated the model on BREAST CANCER and COLON CANCER data with small image sizes, making it computationally possible for running their experiments. We can treat Vanilla C2C as a proxy for Ilse et al.'s approach. Instead of using all the patches, Vanilla C2C sample 64 patches and used bag loss to train the model.
> - **Novelty of the approach:** We completely agree with the reviewer that the WSI classification algorithm is an extensively studied area in recent literature, and multiple groups have proposed approaches tackling this problem. We have included all the related work that has inspired our approach. In most of the recent works, a 2-stage modeling approach has been proposed, whereas there are limited end-to-end approaches given the unique challenges of WSI modeling. C2C is our attempt to contribute to the end-to-end modeling paradigm.

---

### Official Review · AnonReviewer2 · 2021-03-09

**Confidence:** 5
**Preliminary Rating:** 3
**Recommendation:** Poster

**Summary:**

This study presents an end-to-end framework that clusters the patches from a whole slide images into k-groups, and subsampled patches from each group of training is being conducted, and slice level prediction is performed based on attention mechanism. The whole system is called Cluster to Conquer (C2C). Basically,  discriminative training is done based on clustering approach. This is a frequently visited topic in other computer vision and machine learning applications; therefore it is not entirely novel from mythological view point, but application wise it may have a merit.





**Strengths:**

--clearly written paper, motivation is solid, results are promising
-- a good application study
-- MIL with defined costs seem to perform better than the state of the art
-- Any unsupervised algorithm to re-organize data (roughly) and put it into the training will improve the results (in machine learning field this is known, but maybe not tried in this histopathology approach yet, therefore application wise it can be considered as a new too)



**Weaknesses:**

-- The difference between the state of the art Campanella approach and C2C with patch loss is very little, not significant. Vanilla C2C is inferior to Campanella baseline results. Therefore, the main jump in the results are coming from KLD loss, it seems. However, the motivation for choosing KLD loss and detailed demonstrations are missing about the choice and the results, figure 3 is not informative, a better figure c necessary.

--Is not that LeNet5 too primitive to choose? why not deeper and better encoders to model this problem?

--ablation study does not show the benefit of attention aggregation

**Deanonymize Review:**

no

**Justification Of The Preliminary Rating:**

++ results are promising although there are some unclear components still.
++ The paper i s written clearly and the application is important.
-- There are minor to moderate weaknesses about the choice of loss functions and the network architecture,




**Paper Type:**

validation/application paper

**Questions To Address In The Rebuttal:**

--Related to weakness mentioned above about KLD loss, figure 3 c and figure 3 in general is not informative, and there is no solid background and motivation for using KLD loss, its detailed explanations are missing why and how it affected the results so much while most of the results with vanilla baselines and other loss functions are not that different (very close to each other).

--LeNet5 is a primitive choice, what happens if more advanced encoders are being used?

--ablation study does not show the benefit of attention aggregation

--any chance to compare with LI et al 2020's paper where self-supervised contrastive learning is use ?

**Special Issue:**

no

---

> ### Author Response · Authors · 2021-03-18
> **Response to Reviewer2**
>
> Thank you for your valuable feedback and comments!
>
> Kindly find below our response to the comments -
>
> - **KLD loss:** We included KLD loss as It helps our framework in regularizing the attention distribution for the same cluster patches. Without KLD loss, the attention module picks up a set of patches during training and gives high attention to them, ignoring other relevant patches even if it falls in the same cluster. Further, KLD loss also regularizes the attention distribution for all the diseased/ positive class patches. As we demonstrate in the MNIT bag task and Figure 2, when we design an MNIST bag task in which a bag of MNIST numbers is positive if it contains instances of 8, 9, or both. On training with bag or instance loss, the attention module randomly picks up either instance of 8 or 9 and gives it high attention ignoring the other. In contrast, when we include KLD loss, it uniformly distributes the weight to both the classes (8 and 9). We have updated Figures comparing different MNIST bag testing scenarios and have added more details in the method section for explaining the motivation for including KLD loss.
> - **LeNet5 encoder:** We used LeNet5 in the paper to demonstrate the impact of different loss terms on the attention weight distribution. As LeNet5 perfectly learned the MNIST bag task and gave 100% accuracy, we did not change the architecture and followed the similar problem set-up used in Ilse et al.. Our objective was to demonstrate that independent of the encoder's complexity, how the inclusion of different loss terms affects the attention distribution. It can be seen in Figure 2, the inclusion of KL-divergence loss regularizes the attention distribution for positive clusters.
> - **Comparison with Campanella MIL:** We agree that the performance of Vanilla C2C is similar to Campanella MIL, but we would like to point out that as Campanella MIL approach is max pooling patches to predict WSI prediction, and Campanella MIL-RNN approach use RNN with top-k patches for WSI prediction. The performance tends to fluctuate when the dataset is of small size. We hypothesize because of the same reason, Campanella MIL-RNN is not performing as well as Campanella MIL which isn’t expected behavior in the big dataset (>10,000 WSIs). In comparison, our approach is more stable when the dataset is small. We ran all our experiments three times to verify this and have reported the average performance.
> - **Attention aggregation:** We included the attention module as, along with improving the performance of the model, it also helps us in ranking the contribution of each patch for final prediction. In Appendix Figure 5, we have shown that the high attentive patches overlap with the tumor area for the CAMELYON task providing us with a weak segmentation mask. We have added the table comparing the mean pooling approach against the attention pooling approach for aggregation.
> - **Comparison with [Li et al. 2020](https://arxiv.org/pdf/2011.08939v1.pdf):** Yes, we can compare our approach with the proposed self-supervised contrastive learning-inspired DSMIL approach. In the paper, the authors have reported for single scale WSI images; they achieve a 0.8944 AUC score on CAMELYON 16 dataset, whereas we are achieving a 0.9112 AUC score. Further, we plan to conduct a study to compare how pre-training the backbone encoder using self-supervised learning, multi-task learning on different histopathology tasks performs against imagenet training with our C2C architecture in our follow-up work.

---

### Author Response · Authors · 2021-03-18
**General Response to Reviewer**

We want to Thank all the reviewers for their valuable feedback and comments!

All of the comments have helped us in improving our paper. We have addressed each of the comments separately and hope it clarifies the concerns raised.

Following changes were made to the draft based on all the inputs:
- **Figure:** Updated the images and their captions to high quality.
- **Motivation for KLD loss:** Included in the method section 3.4.
- **UMAP Plot:** Added UMAP plot in the Appendix for the duodenal biopsy classification problem to demonstrate WSI representation separation.
- **Sensitivity analysis of Gamma:** Included a table in the Appendix to highlight the model's performance for different weights of KLD-Loss.
- **Results Update:** To further compare the performance variation in Table 1 that can occur due to random sampling of patches and different combinations of weights, we ran all of the experiments reported in Table 1 three times and have reported the average performance.
-**Updates to Related work:**
    * MIL and Weak Supervised Learning: Added reference to literature linking multiple instance learning to weakly supervised learning approaches for WSI classification.
    * CLAM: Included CLAM reference as a 2-stage approach for WSI classification.
- **Mean Pooling vs. Attention Pooling:** Added table in the Appendix for comparing the performance.
- **Time Complexity:** Added table in the Appendix for comparing inference time per WSI in duodenal biopsy test data.
- **Symbols in the equation:** Included explanation of all the symbols.
- **Maximum number of patches = 64:** Added justification for the maximum number of patches
- **Code:** Open-sourced implementation available at https://github.com/YashSharma/C2C.

---

### Meta-Review · Area_Chair1 · 2021-02-22

**Recommendation:** Accept (Poster)

**Metareview:**

The paper proposes a framework for classification of whole slides images, where the DNN essentially relies on patch processing (as is the norm). The paper proposes an aggregation module that relies on a DNN to perform a weighted average of the patch representations (Section 3.2). The paper claims end-to-end learning, but this involves a sampling step that hasn't been shown to be suitable for the purpose.


**Paper Type:**

validation/application paper

---

### Decision · Program_Chairs · 2021-03-31

Accept